# Vague Pension Future: Empirical Evidence from the Israeli Radical Privatized Market

**Ishay Wolf \* and Smadar Levi** 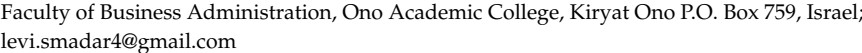

Faculty of Business Administration, Ono Academic College, Kiryat Ono P.O. Box 759, Israel;
levi.smadar4@gmail.com
\* Correspondence: ishay.wo@ono.ac.il

**Abstract:** We examine the future benefits of the Israeli privatized pension system, which is considered as a model of transition to funded pension systems worldwide. This research is based on an extensive database obtained from one of the largest traditional private funds in the market. The results paint a concerning picture regarding the adequacy of benefits and quality of life in old age. Israel's radical privatized pension model signals a warning to other nations. We show that, even with high returns, most individuals cannot handle the magnitude of financial and labor risks accumulated during their career and retirement. We recommend more balanced government intervention as well as the use of risk-sharing mechanisms such as providing minimum pension guarantee and strengthening the unfunded social security pillar.

**Keywords:** funded pension scheme; social security; poverty; minimum pension guarantee; public policy; Israel

## 1. Introduction

Aging societies are a growing challenge to public pensions, particularly pay-as-you-go (PAYG) systems, in which the working population finances the pensions of the retired. International organizations and national policy experts recommend a shift to a multi-pillar pension architecture with more funded elements (Holzmann and Hinz 2005; Lurie et al. 2021). This may involve replacing traditional public schemes with individual pension accounts. Governments aiming to hedge fiscal risks often adopt this design in the belief that an aging population will unavoidably lead to the collapse of PAYG and other non-contributory systems (Madero-Cabib et al. 2021).

While there has been a clear trend toward the financialization of pension policy in recent decades, reforms have not led to convergence among OECD countries (Lever and Michielsen 2016). Differences in the regulation of pension funds are responsible for varying degrees of individual financial risk. In some cases, this has led to what is known as "the financialization of daily life" (Hassel et al. 2019).

One result of this privatization trend is the transfer of financial risks from the government to the individual. The rise of defined contribution (DC) pension plans has created a direct link between the level of benefits and financial market fluctuations for many households, which have effectively become "everyday" investors (Langley 2008). Consequently, as participants become more actively involved in their finances, there is a greater need for risk managing, hedging, savings, and choice of investment strategies for individuals.

The financial crisis of 2008 and the more recent COVID-19 pandemic have demonstrated the sensitivity of old-age benefits not only to capital market volatility (Lever and Michielsen 2016), but also to changes in wage levels and fluctuations in contributions. As a result, people's trust in the expected pension benefits has been shattered (Gierusz et al. 2022).

The increase of financial risks associated with frictions in the labor markets, as well as global systemic risk, provide an opportunity to investigate correlated shocks on pension benefits and contributions. Notably, those workers who stand to benefit from recent global pension reforms implemented by many Western countries are only expected to retire in five to 10 years (Bonenkamp et al. 2016). Current literature recognizes an educational gap of knowledge on how to manage personal pension accounts (Yong et al. 2018). Little has been accomplished to consider the outcome for those people planning to retire further in the future.

Another major area of uncertainty is the socio-economic effect of privately funded pensions. A number of recent papers point out that private pensions are associated with higher levels of income inequality and poverty among the elderly (Been et al. 2017). Van Vliet et al. (2012) hypothesize that those pension reforms that reduce public funding components have led to higher levels of income inequality and poverty among the elderly. Recent research shows that 33% of employees will be either poor or near-poor when they retire, and 55% will rely only on their social security income (Rubinstein-Levi 2021).

This empirical research aims in particular to determine the adequacy and sustainability of such funded pension arrangements worldwide. Indeed, recent scholars argue that the sustainability of a pension system depends on its benefit adequacy and strength when faced with economic shocks or significant financial risks (Ebbinghaus 2021; Wolf and Lopez Del Rio 2021).

We find the Israeli pension market to be a suitable "laboratory" for analyzing changes to a society's primary pension system. Israel is a small, Western, open, and developed country, which has undergone a radical transition from a dominant PAYG-defined benefit (DB) scheme to an almost purely capital-funded model with little government intervention, based on individual accounts.

Based on a unique database of 10,000 real-pension records from October 2021, we simulate the future pension benefits of current participants in privatized capitalized individual accounts pension plans. The records database was obtained from a corporation with the largest private pension fund in Israel—'HL. Insurance'. According to Amaglobeli et al. (2019), the evolution of the pension market in Israel reflects the benefits and other changes seen in other Western countries with dominant funded pension schemes. Hence, we feel comfortable in applying our conclusions and suggestions from this research to other countries, considering or implementing pension reforms.

Most of the countries that undertook pension reforms in a wave in the 1990s have reverted back their pension scheme to DB PAYG (Ortiz et al. 2018). However, "Israel's pension system remains stable, with consistent capitalization trends" (Giorno and Adda 2016, p. 2). We count the Netherlands, Iceland, Australia, and Switzerland as having similarly dominant stable funded and private pension schemes (Hassel et al. 2019).

We aim to delve into the future consequences of major pension reform toward capitalized-funded pension schemes, as a contribution to the debate on the proper balance of government intervention in the pension market. We assert that a transition solely to funded pension schemes does not provide the full potential to retirement benefits that provide dignity in old age and even puts participants at risk of poverty and severe low income in their later years, despite a lifetime of hard work and savings. We point out the system's vulnerability to financial and systemic shocks. These may include periods of unemployment, low market returns, wage cuts, and early withdrawal from pension accumulation.

Moreover, we show that Israel's pension system provides less room for redistribution and leaves individuals exposed to substantial income inequality and market fluctuations. We learn that even in a stable market yielding high returns, there is a strong case for a substantive, unfunded pension pillar in a participant's old-age portfolio. These findings are consistent with the literature on ways of preserving a proper balance of unfunded pillars in the pension system (Wolf and Caridad Ocerin 2021; Beetsma and Bovenberg 2007).

In addition, we find that the financialization of pensions is associated with socioeconomic questions of income inequality and poverty. A privatized system amplifies income inequality through an accumulative effect on future generations. These social considerations must come as key principles when building the pension system. What fits one market in a specific country might not be the answer in another one (Barr and Diamond 2009).

We link that topic to another major area of debate: pension savings. People tend to save insufficiently for retirement even in markets with relatively high contribution rates and high market yields (Endblad et al. 2021). This socio-economic problem heightens the risk of more widespread poverty among the elderly and greater demands for various forms of government social assistance, increasing the overall likelihood of fiscal imbalances. We find it important to have a set of meaningful measures and to diagnose the extent and nature of any inadequacy in retirement savings. It is also important to develop effective means to assist individuals to implement appropriate savings targets, for example by communicating clearly to people the need to take action and by adjusting the institutional environment accordingly (Amaglobeli et al. 2019).

This research also finds a lack of knowledge among the Israeli population about pension deployment after retirement. Most people do not know how to optimize their behavior to ensure adequate benefits. Financial education may help encourage individuals to be more active in their portfolios and deployment of their pensions when the day comes.

Globally, one may determine that the first significant wave of participants affected by the significant pension reforms during the 1990s has not yet retired and will do so in the coming years. Only a few studies have been conducted on the Israeli pension system. Most of the researchers, such as Benish et al. (2016) and Menahem Carmi and Kimhi (2018), have dealt with the influence of management administrative costs on individual accounts.

To the best of our knowledge, this is one of the few studies to encompass empirical types of research, analyzing funded pension benefits based on real wage and accumulation records. Many other papers try to simulate the expected benefits in funded pension designs based on theoretical models. For example, Wolf and Caridad Ocerin (2021) determine the optimal portfolio or pillar size in retirement. Some scholars have researched funded pension schemes from a risk-sharing perspective. For example, Chen et al. (2014) link the funded scheme to debt and tax policy. Bonenkamp et al. (2016) research pension reforms in different countries and how these reforms advance the participants' welfare in an aging environment. Another strand of literature measures the adequacy and sustainability of local pension systems; we might mention here Fajnzylber (2019), who has analyzed the Croatian pension system. Knoef et al. (2016) have also measured retirement savings based on national data from the Dutch system. Meanwhile, Burnett et al. (2018) explore savings trends in the Australian pension system.

The remainder of this paper is organized as follows. In Section 2, we describe briefly the Israeli pension system and its evolution. We also provide a comparative perspective of its pension system and its radical privatization profile. Section 3 provides a description of the old-age benefits model based on real empirical data.

After outlining the methodology used in this study in Section 4, we discuss the main insights from the model results in Section 5. We further explore the implications of the results in Section 6 and suggest some recommendations to improve the adequacy and sustainability of the funded pension system, looking specifically at the Israeli example. Section 7 concludes the study.

## 2. Pension System Description

### 2.1. The Israeli Experience

After a series of radical reforms, the pension market in Israel is completely capitalized and privatized, which frees the government from the potential of longevity risk and fiscal burden. This has been bolstered by impressive macroeconomic results in the Israeli market in the past 35 years.

With almost full coverage, the new pension funds are private defined contribution (DC) funds, in which, upon retirement, an individual's benefits are determined by the amount of money accumulated in their account. This accumulation is a function of the market return as well as monthly contributions from a person's wages throughout their entire career.

In the pre-reform era, occupational pensions included a PAYG DB scheme controlled by the unions, as well as a governmental lifetime annuity plan for soldiers and public servants. Pension savings were not obligatory, as only public sector workers and employees under sectoral agreements were enrolled automatically. In these schemes, upon retirement, the individual received a guaranteed monthly pension equal to a predetermined percentage of their salary. All funded instruments were invested solely in designated governmental bonds.

This limited governmental DB pension scheme was considered generous and lacked competition. For these reasons, the old funds fell into an actuarial deficit that amounted to about 36 billion dollars. This deficit was, in fact, the background to profound stabilizing reform in the mid-1990s, including capitalization actions, with the enforced support of the labor organizations.

As of 1995, pensionary coverage stood at only 61% of the employed working population. According to Lurie et al. (2021): "In 2002, pensionary coverage rose slightly, reaching 73% of employed workers, but about 50% of the adult population in Israel lacked pensionary insurance. The rate of the insured was particularly low among women, minorities, part-time workers, and low-paid workers." There was also a low rate of insured among the self-employed. Consistent with a global trend, a mandatory pension for the self-employed was instituted by law in 2008. Nowadays, overall pension coverage is quite high, accounting for 85.3% of Israeli employees (OECD 2021).

In 1995, new participants were obligated to save through capitalized funded individual accounts. In 2003, the old-age allowance linkage to the average wage was abandoned. It was linked instead to the Consumer Price Index, thus halting the gradual increase in the actual value of the allowance and reducing fiscal expenditures. In the same year, in another step toward pension market capitalization, the coverage rate of governmental designated bonds, with high returns, was reduced to 30% of pension-accrued portfolios. Since 2017, the mandatory investment in these bonds ("Arad") has been raised to 60% for savers above the age of 60, to 30% for those in the 50–60 age range, while the allocation to savers under 50 years of age is being gradually reduced to zero (Wolf et al. 2021).

The Israeli government instituted a mandatory pension law to increase the rate of pension savers in the population, effective on 1 January 2008. That increased coverage within a decade from 35% to 78%. A decade after that, contributions became compulsory also for the self-employed.

Consistent with its historical policy of minimum intervention, in 2022 the government stopped issuing designated bonds with a high rate of return. In the past, designated bonds incorporated in the pension portfolio were similar to a guarantee of a minimum rate of return.

### 2.2. A Comparative View

From a global perspective, the mixed pension pillars system is increasingly common, with different models representing a variety of government responsibilities or interventions.

A popular mixed pension scheme is the notional defined contribution (NDC) or "point" system, which provides pension benefits according to wage path and working period (Devolder et al. 2021). The government manages these funds and determines the yearly rate of returns or growth rate of accumulations. Among the countries that have implemented such mixed pension schemes, we can mention Germany, Poland, Estonia, Italy, France, and Sweden. Other mixed designs involve both funded and unfunded pension pillars with a varying magnitude of public pension spending (see Panel A in Figure 1). Examples include France and Switzerland.

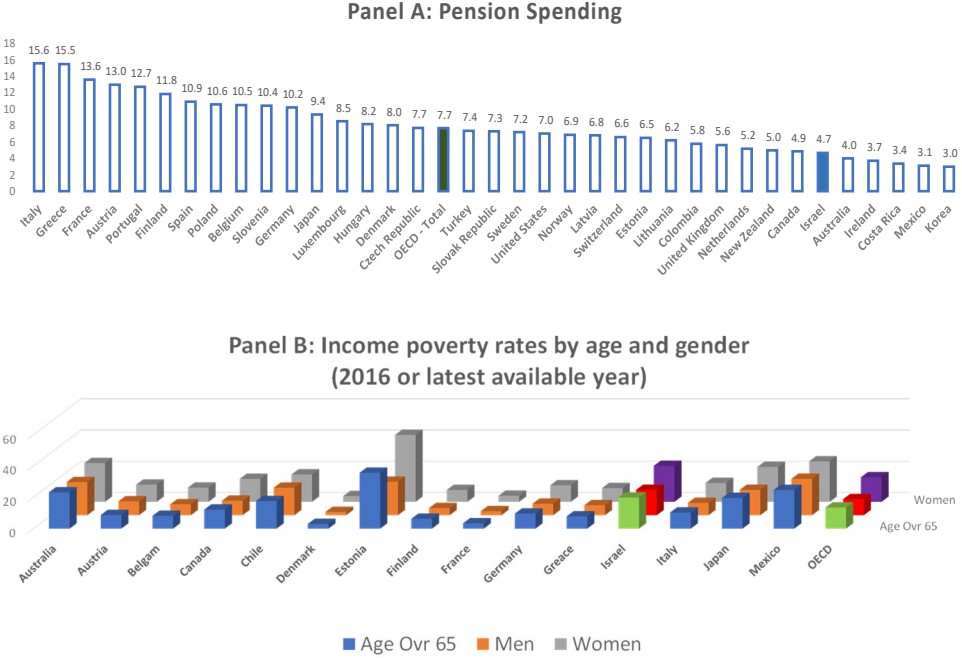

**Figure 1.** Poverty and Pension Spending in international Comparison. Authors' elaboration based on the OECD data, 2020.

### 3. The Three Pillars: A Pension Model

Israel belongs to a growing group of countries that privatized their pension system via for-profit financial bodies. Among this group, one can find the United States, Denmark, Switzerland, the United Kingdom, the Netherlands, Chile, Canada, Australia, Ireland, Iceland, and Mexico. This pension design includes a dominant funded pension design but also a developed complementary capital saving in individual accounts, which is often called the 'third pillar' (Orenstein 2008).

Israel's pension reform in the 1990s can be compared to those introduced in several Latin American countries, starting with Chile in the early 1980s, as well as several CEE countries such as Hungary and Slovakia, which adopted the extreme view of privatizing national pensions according to a purely DC model (Kay 2014).

The difference is that Israel has kept to the principle of capitalization, while most of these countries have rolled back their pension systems toward more significant government intervention. Indeed, Israel's public expenditure on pensions has decreased constantly and is now one of the lowest among the OECD members, at 4.7% compared with the average OECD figure of 7.7% (see Figure 1 Panel A). This is reflected by the diminished size of the universal public first pillar. The allowance of social security is very low compared with other Western societies, at about only 420 USD a month on average.

The pension contribution rate in Israel is close to the average OECD member. As in most other countries, low-earning participants are exposed to a lack of contribution from their employer or may rely on income on the black market (Rubinstein-Levi 2021). That may explain the incomplete pension coverage in Israel and other Western countries that have implemented mandatory contributions.

Thanks to the development of capital markets and growing competition, administrative fees in Israel are slowly decreasing. They are still substantive, however, "double those in the U.S., where large pension funds collect, on average, 0.15% of the accrued reserve and 0.42% of the contributions" (Rubinstein-Levi 2021, p. 4). On the other hand, Israeli participants benefit from a high yield that reaches on average more than 5.5% a year, compared with the OECD average over the past decade, which was around 2.5% (OECD 2021).

Traditionally, wage differences among demographic groups, successive waves of immigration, high real-estate prices, and the growth of the hi-tech industry are among the reasons for high degrees of income inequality in Israel. The top two earning deciles together receive 44% of the total wage revenue. The remaining working population combined receives 56%. It should be noted that this income distribution has hardly changed in the past two decades (Giorno and Adda 2016). The funded pension scheme only intensifies economic gaps among earning cohorts and does not contribute to income redistribution. The twofold reason is, on the one hand, the accumulation-accrued effects and, on the other hand, differences in the ability to hedge risks among earning cohorts.

In this research, we point out that the socio-economic strength of the country affects its ability and motivation to balance the pension system with more government intervention and unfunded vehicles. According to Figure 1, Panel B, the poverty rate in Israel (19.9%) is still high compared with the OECD average (13.5%). In old age, 50.3% of the elderly population lives below the poverty line. Some 23% of women live in poverty, compared with only 15.5% in the OECD on average. The high poverty rate exacerbates the high-income inequality level, which exceeds by a quarter the OECD average (Bleikh 2016).

### 3.1. Wage Path

We obtain real wage values of 10,000 random participants as of October 2021. These records enable us to calculate pension accumulation at retirement.

First, we simulate the expected wage path for each individual until their retirement. We assume the wage path is a function of GDP per capita, which varies as a function of earning decile. High earners benefit from a higher periodical share of the GDP per capita growth rate. In addition, we assume a life-cycle earning profile, whereby a person's wage grows differently at different periods in their career. The member, wage development can be described as follows:

$$W_{i,t+1} = W_{i,t} * (E_t a_i) \, g_t \tag{1}$$

where $W_{i,t}$ represents the wage in the year $t$. $E_t$ stands for the earning decile, indexing from the GDP per capita. $a_i$ represents the age indexing factor.

### 3.2. Public Social Security

This social security benefit is a function of residency and the neglected influence of wage insurance. This pillar is with only 5.3% of GDP, while the average OECD rate stands at 10% (NII Financial Report to 2020). Based on the NII rules described above, the maximum amount for an individual in 2021 stood at NIS 2300 a month.

Assuming a constant debt/GDP ratio, in each period the total working population's contributions are equal to total benefits payments to retirees. One may imply that the public benefit is determined by the following rule:

$$\tau^U \overline{W}_{t+1} N_{t+1} = p^U_{t,T_R} N_t \tag{2}$$

where, $\tau^U$ is the contribution rate to the unfunded pension pillar. $p^U_{t,T_R}$ is the unfunded pension benefits paid to generation $t$ in the period of $t+1$. $N_t$ is the generation size at time $t$.

### 3.3. The Funded Pension Scheme

The capitalized expected pension benefit depends on a participant's accumulations throughout their working career. The employee and the employer together contribute the fraction $\gamma \tau w_t$ from the insured salary. Pension accumulations are invested in free market securities. The fund deducts insurance premiums $\varnothing$ from contributions for spousal benefits and disability risk. Likewise, the fund charges an administration fee of M% from the contributions.

The funded accumulations earn the annual rate of return, $r_t$. The return is no longer includes investment in designated bonds with the constant return. This rate of return also follows a "Brownian motion" in the following way:

$$dr_t = \mu_r dt + \sigma_r B^A dt \tag{3}$$

Here, $\sigma_r$ is a constant standard deviation. The first phrase is constant drift, and the second is volatility drift, respectively, from left to right. The funded benefit can be describes as:

$$p^F_{t+1} = \frac{\left[ A^F_t + (1 - \varnothing)\gamma\tau \sum_{s=t}^{T_R} W_{t,s}(1 + r_t - M)^{T_R - t} - \varepsilon_t \right]}{\theta} \tag{4}$$

Here $A^F_t$ is the participant's balance at time $t$, $\varepsilon_t$ is the personal withdrawal amount from the severance pay component in year $t$. We conduct stress tests on $A^F_t$ and $r_t$, examining shock in the labor and capital markets.

$\theta$ generally represents actuarily the effective average number of months at retirement and is a function of a participant's gender, retirement age, and the market return. The annuitization mechanism of the private benefits includes longevity insurance, as the allowances are paid until death. This system generates redistribution from those who die early to the elderly who stay alive beyond the average age of death.

### 3.4. The Third Pillar

Based on the unique database, we also obtained savings data under the third pillar. That includes private contributions to long-term savings funds, which benefit from generous tax exemptions. Although contributions to this third pillar are not mandatory, contributions to these funds are very popular in the West and are another financial device to increase savings toward retirement. We simulate the present empirical accumulations to the retirement age using the same method as the second pillar, above.

### 3.5. Minimum Pension Guarantee

Meeting the European Commission's 2020 target for adequacy old age pensions, and avoiding poverty and high-income inequality levels, many governments implement unfunded mechanisms as part of a mixed system (Cantillon and Lynch 2017). we consider the potential effect of incorporating a flat minimum benefit guarantee at the poverty line level. We also measure the cost burden of implementing the guarantee and its fiscal weight.

### 4. Methodology

We simulate wage and pension benefits based on the records obtained, which are updated to October 2021. Each record line includes the participant's age, gender, family status, current insured wage, current balance in the second and third pillars, and private pension actuarial coefficient.

The research is limited to financial information from the sampled fund. Participants may have other savings in other funds. However, that is generally true for savings in the third pillar and is less the case for private pension accounts. Hence, we are cautious about drawing conclusions on the third pillar's savings and mainly focus on the pension accounts.

From the data draw point of October 2021 and based on predetermined wage development assumptions, we simulate the expected wage path for each participant until retirement, rolling his wage in a cyclical model (see for example Endblad et al. 2021). naturally, as the participant is older, we had to roll for fewer years their wage until retirement and hence the financial outcome is less doubtful. As the participant is younger, clearly labor and financial risks will realize during his career. From that the assumed wage path, we derive the future pension contributions, returns and balance and benefits for each sample participant. Accordingly, and as detailed in the previous section, we analyze these future

benefits according to age cohorts and earning deciles. We consider two correlated risk variables that affect the wage path and pension benefits: GDP per capita, and the private pension return. These two factors influence the participants' wages and social security allowances.

Private pension allowances have generous tax exemptions. Accordingly, and due to the high diversity in these withdrawals, we found it impossible to consider the net benefits through this model. We, therefore, provide the benefits in gross terms.

We have calibrated the simulation based on the Israeli market, which we find to correlate closely with the average values across the OECD. We set an average annual net (after administrative costs) private pension fund return of 4%.

## 5. Results and Insights

Before delving into pension performances, one has to recognize the limits of the private pension model. Although its prevalence is relatively high in the Israeli market, 18% of the population still lacks any private pension coverage. According to Bonizzi and Guevara (2019), this rate corresponds to other Western economies implementing private pension systems.

Naturally, pension samples cannot reflect the uncovered household. Low-earning deciles are not capable of saving enough for pensions and instead lean on the universal social security allowance, where it is available. At the other end of the income scale, high-earning individuals typically invest in other financial assets in addition to compulsory pension contributions. Table 1 provides the sample's monthly wage distribution, which we find to be close to the total population data as reported by the OECD (OECD 2021). In addition, we can point to a high level of income inequality. This itncludes a concerning average wage gap of nearly 25% between men and women.

**Table 1.** Sample's wage distribution (USD).

| Category | | Average | Men | Women | Difference |
|---|---|---|---|---|---|
| <35 | Total | 3029 | - | - | |
| | Married | 2974 | 3529 | 2633 | 25.4% |
| | Divorces | 2935 | 3267 | 2803 | 14.2% |
| | Single | 3057 | 3460 | 2643 | 23.6% |
| 35–45 | Total | 3585 | - | - | |
| | Married | 3595 | 4101 | 3173 | 22.6% |
| | Divorces | 3094 | 3464 | 2923 | 15.6% |
| | Single | 3656 | 4070 | 3243 | 20.3% |
| | Widow | 3203 | - | 3203 | |
| 46–55 | Total | 3280 | - | - | |
| | Married | 3335 | 3885 | 2854 | 26.5% |
| | Divorces | 2998 | 3908 | 2709 | 30.7% |
| | Single | 3171 | 3355 | 3003 | 10.5% |
| | Widow | 2613 | 3031 | 2574 | 15.1% |
| 56–67 | Total | 2741 | - | - | |
| | Married | 2793 | 3275 | 2353 | 28.2% |
| | Divorces | 2563 | 3198 | 2333 | 27.0% |
| | Single | 2709 | 2577 | 2783 | −8.0% |
| | Widow | 2249 | 3332 | 2033 | 39.0% |

Source: Own elaboration.

The life cycle income from the sample is close to that reported from the literature, with a peak in the late 40s and some reduction toward retirement (Ejrnæs and Jorgensen 2020; Rigg and Sefton 2006).

### 5.1. Benefit Adequacy

Table 2 summarizes the expected old age funded benefits as a function of age cohorts and family status. It can be easily noticed that the funded privatized revolution over the past three decades has penetrated slowly to the working population and is realized in higher coverage among the younger generations. Although the performances for the younger section of the population are stable and encouraging, a concerning picture arises regarding the transition generation, representing those who are above the age of 50 and have not consistently contributed to a personal account over their career.

**Table 2.** Expected Monthly Funded Benefits (USD).

| Category | | Average | Men | Women | Difference |
|---|---|---|---|---|---|
| <35 | Total | 4468 | 5530 | 3552 | 35.8% |
| | Married | 3886 | 4993 | 3206 | 35.8% |
| | Divorces | 3867 | 4990 | 3417 | 31.5% |
| | Single | 4758 | 5729 | 3764 | 34.3% |
| 35–45 | Total | 2994 | 3710 | 2391 | 35.5% |
| | Married | 2951 | 3647 | 2372 | 35.0% |
| | Divorces | 2284 | 2819 | 2034 | 27.8% |
| | Single | 3291 | 4014 | 2570 | 36.0% |
| 46–55 | Total | 1594 | 2040 | 1240 | 39.2% |
| | Married | 1640 | 2073 | 1262 | 39.1% |
| | Divorces | 1298 | 2021 | 1068 | 47.2% |
| | Single | 1575 | 1809 | 1361 | 24.7% |
| | Widow | 892 | 1594 | 828 | 48.1% |
| 56–67 | Total | 694 | 893 | 542 | 39.3% |
| | Married | 722 | 912 | 548 | 39.9% |
| | Divorces | 597 | 796 | 525 | 34.1% |
| | Single | 656 | 721 | 620 | 14.0% |
| | Widow | 486 | 1045 | 374 | 64.2% |

Source: Authors' calculations.

Our findings regarding the expected funded benefits of the 56–67 age group are somewhat concerning. According to Figure 2, the expected benefits of this age group from the first and the second pillars are below the poverty line. For women, these forecast benefits are even worse, reaching the levels of universal social security allowances. However, we report another expected benefit of 330 USD a month to this age cohort, which in average enables total benefits a little above the poverty line. The third pillar includes a high volatility.

The data support anecdotal reports of late enrolment in the funded pension system. We can infer that some of the participants within this age cohort are still enrolled in the former DB pension scheme. Others, naturally, possess other financial assets, such as real estate or other savings under the third and fourth pillars described in the previous section. It is quite challenging to track the financial position of this age group. Even so, we can expect a large portion of that age cohort to be facing poverty in retirement. As a consequence, the

government will presumably need a temporary but substantial fiscal outlay to ensure this transition generation is adequately provided for.

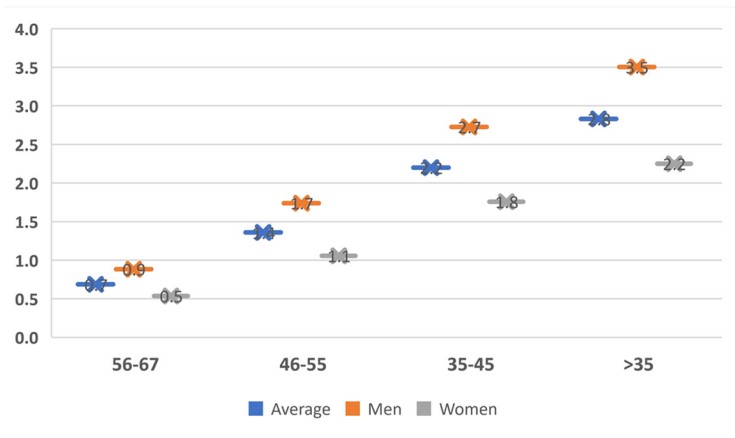

**Figure 2.** Expected Private Old-age Benefits in Terms of Forecast Periodical Poverty Line. Source: Authors' calculations.

For the younger age group of 46–55, the replacement rate is around 50% of the average wage, which falls close to the average corresponding value in the OECD countries. In younger age cohorts, the expected benefits are high, proving the main idea of high expected benefits with consistent contributions.

Even so, the funded system cannot alleviate socio-economic problems. Rather, it is likely to amplify them. The high-income inequality levels seen during the working phase will only be increased during retirement. The current system does not provide a cushion against low benefits, even in cases of consistent contributions during decades.

"The growing predominance of the second pillar, to the cost of the first pillar, may cause pension gender gaps and other inequalities in Israel to widen" (Bowers and Fuchs 2016, p. 3). Figure 3 plots graphically the increasing differences in private benefits between men and women. One can thereby deduce the funded pension scheme will exacerbate current socio-economic challenges. If, during the working phase, the wage differences between men and women were around 25%, then depending on age cohort the difference in benefits in retirement are expected to widen to about 35%. Larsen et al. (2022) explain that some of these gender differences may be due to women investing larger amounts into less risky, but potentially less rewarding, pension products compared to men.

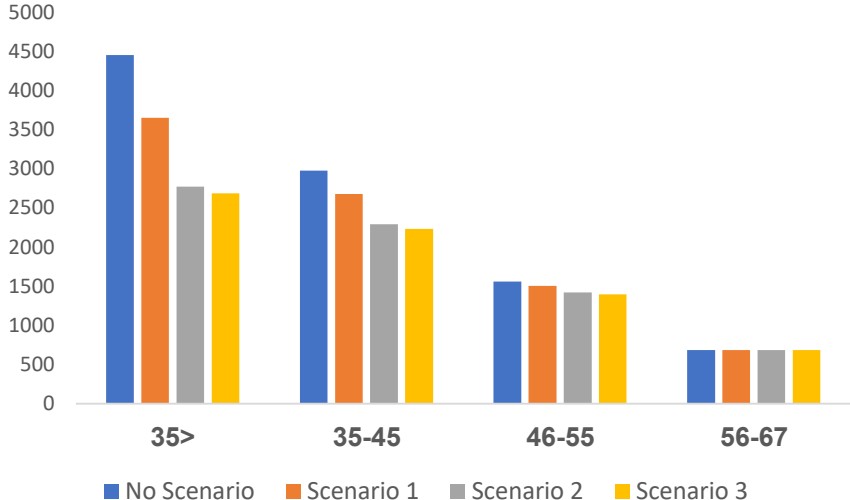

**Figure 3.** Pension Benefits by Commutative Stress Tests.

Traditionally, wage differences among demographic groups, as well as high real-estate prices and the successes of Israel's hi-tech industry, are among the reasons for high-income inequality levels. However, the funded pension with the accrual effect only intensifies economic gaps among earning cohorts and does not contribute to income redistribution. If the central planner attends to address this matter, a higher share of the unfunded pillar should be considered. We add to this the need for financial education in managing the individual account in life-time perspective, which is lacked in low earning deciles.

The third pillar thus becomes an important saving vehicle, increasing the adequacy of pension benefits in old age. Based on provident funds and advanced study of funds managed by the same private organization we can assess that third pillar accumulations amount to about 25–33% of pension accumulations.

One problem arises from the high volatility among participants contributing to this pillar. They are typically high earners, which amplifies the overall societal problem of income inequality in old age. We note that up to the sixth earning decile, the magnitude of the savings under this third pillar are not significant. The volatility of savings is also realized by marked differences in savings between men and women, which sums on average by 30%. Although gender benefit differences are not the scope of this paper, this financial data implies substantial differences in outcomes for men and women in old age.

*5.2. Vulnerability to Labor and Capital Shocks*

Similar to the COVID-19 economic crisis effect, we examine the joint influence of risks in the labor and capital markets on old-age benefits. When constructing a simulation of an individual's entire career, we consider both cyclical difficulties in the labor and capital markets, which may influence the individual accumulation for retirement. These stress tests were formulated from real scenarios in the Israeli market during the past 20 months. The scenarios' effects are detailed below and summarized in Figure 3.

Market Fall: According to the simulations, annual market falls of 10% affect the expected funded benefit by 4% on average. Hence, assuming the realistic scenario of a cyclical market fall every 10 years or so implies a 12% reduction to the younger cohort or fall of $840 a month. These traces will be minimized if the market recovers and compensates for those losses. However, for those over 50, it is not straightforward to assume a full recovery above the average yield so some of the market falls will be absorbed by their funded benefits.

Unemployment: Periods of unemployment are first realized by a shortfall in contributions to the pension account. We find that, on average, a month's hiatus in contributions is equivalent to a 1% reduction in old-age funded benefits. When the unemployment period is earlier in a career, naturally its impact increases because of the accrued effect.

Withdrawal: We add some liquidation of severance pay for different reasons, such as housing, marriage, childcare, and preservation of life quality between jobs. We measure the effect of withdrawing 20% of accumulations by another 4% reduction in funded pension benefits.

When both scenarios emerge together, for example in cases of unemployment or large expenditures, the effect on pension benefits can be significant. Indeed, one must notice that periods of unemployment often coincide with a systemic financial crisis. That includes the joint effect of accumulation withdrawal and market falls. In the above assumptions, those three cyclical scenarios might lead to a severe benefit reduction. Confirming Wolf and Caridad, we ascertain that "even short systemic shocks leave traces on future expected benefit levels" (Wolf and Caridad Ocerin 2021, p. 18). That is also true when the participant is younger.

Periodical shocks in the labor and capital markets intensify the urgency of balancing the pension system by using the unfunded pillar. The reason is that the PAYG pension scheme has a weak correlation to funded accumulation (Wolf et al. 2021; Barr and Diamond 2009). Due to the accrual effect and improvements in accumulations over the years, in a

mixed pension system the hedging effect is naturally more effective in old age compared with earlier in life.

### 5.3. Implementing a Minimum Pension Guarantee

As discussed above we assume a minimum pension guarantee in the poverty level. Implementing a public guarantee will be relatively expensive in the next 15 years due to the lack of coverage for the transition generation as per the funded pension scheme described above.

We suggest financing the guarantee using several sources. First, the expected reduction of budgetary pension spending for public servants will provide an available budget from 2033 with a growing line of 0.6 billion NIS. Another source of financing for the guarantee could be derived from redirecting the subsidy inherent in the issuance of the designated bonds. Over the past year, the cost amounted to 7 billion NIS. The alternative cost is expected to grow by 0.8 billion NIS a year. To stay conservative, we assume a constant financial source with no growing availability of public budget. This structural change enables a twofold possibility: for the government to raise public debt at a lower price, and for the participant to benefit from the higher potential yields in the market. Figure 4 depicts the guarantee cost and the alternative economic sources to finance it in a time series.

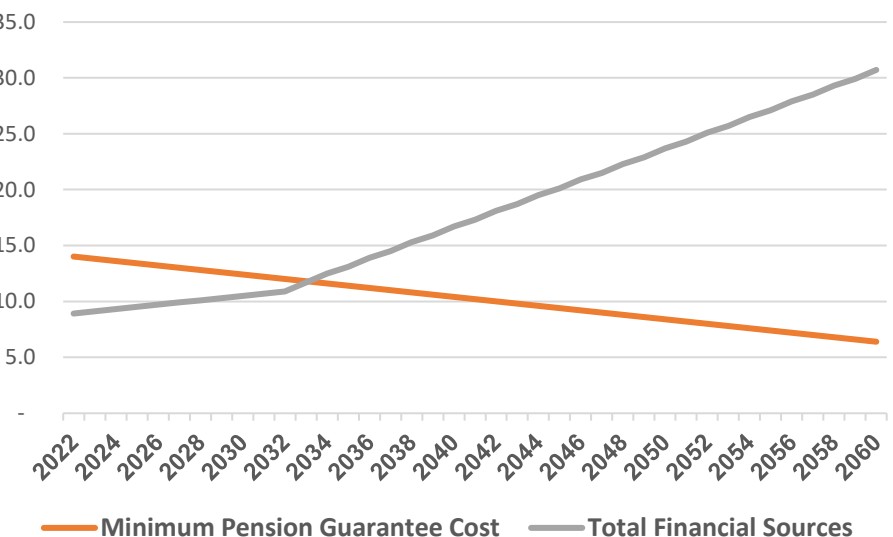

**Figure 4.** Financing Minimum Pension Guarantee at Retirement. Source: Authors' elaborations are based on MOF estimates and the trend of negotiations between the government and security organizations.

Moreover, a minimum pension guarantee is considered as a substitute to other social programs for old age. Hence, these plans are also can be counted as a source of additional financing. According to the NII report, this amount came to almost 1.9 billion NIS in 2020.

### 6. Discussion

Pension saving rate of 28% from monthly wage represent an average rate in the OECD and enables sufficient saving balance for those who persist to work and contribute from early stage in their careers. Pension benefits may not be sufficient to ensure dignity and quality of life in old age due to vulnerability to labor and capital market shocks, as well as likely disruptions to a person's career.

The transition within the pension system represents firstly a transfer of risks from the government, to its individual citizens. Each pension participant is vulnerable to financial risks at different points in their career. For example, at an advanced stage in their working life, market risks tend to carry a heavier weight. By contrast, at a preliminary stage, the risk of unemployment and withdrawal is much more significant. The results further imply a

long-term traces on old-age pension accumulations when the above financial and career risks realize.

A fiscal response is expected to realize for those with low balance and who belongs to the first generation of the funded reform or to new immigrants from the third world countries. The solution for this transition generation may include encouraging savings under the third pension pillar. We believe that the government will have to redirect financial resources to that emerging fiscal problem, although it is not likely to be a permanent challenge. Some of the financial resources available are suggested in the previous section.

Studying from global experience, for example from the CEE countries and Latin America, a funded pension scheme cannot remain sustainable if it lacks adequate benefits which enable honorable consumption in old age. Due to the emergence of financial risks and socio-economic challenges, the entire pension system can be jeopardized, by political pressure to reverse back to the state intervention. As a balancing mechanism, the implementation of an unfunded guarantee can help create a more balanced pension design, or governmental hedging against labor or capital market turmoil. A guarantee significantly improves low-earning replacement rates, increases income redistribution, and reduces poverty in old age. Based on the high degree of old-age poverty and income inequality, we suggest focusing on total benefit adequacy from the first and second pillars, compared with the current rate of return on the capitalized fund.

We suggest that the guarantee be financed by amortizing designated bonds, abolishing means-tested substitute programs, and redirecting the available budget for old budgetary pension expenses. By implementing the guarantee, the underlying asset is changed from the market's return to the participant accumulations, which is much more controlled by the pension actors in the field.

## 7. Conclusions

This paper explores the possible outcomes of a highly funded-capitalized pension system in the context of the average OECD country. We find the Israeli pension market to be a laboratory for studying the funded-capitalized popular global shift toward pension privatization and capitalization. We simulate the pension benefits and allowances that can be expected in old age, based on real pension records.

Given strong U.S influence and a long tradition of a liberal economy, both privatization and capitalization are deeply rooted in the Israeli market, with no reversal of this trend expected in the years to come. However, the results outlined in this study might raise some doubts among central planners regarding the capabilities of participants to manage and handle financial risks for such a long period. They should also focus attention on the need for risk-sharing mechanisms.

We identify a lack of diversified sources for old age pensions, both of which are needed to ensure adequate quality of life at retirement. The transfer of risk from the government to individual citizens must lead to a parallel behavioral change. Each pension participant must be aware of changes to their accounts, and try to minimize periods of insufficent contributions.

The results assert the importance of consistent contributions along the career life path. For low earners, it might not be enough to move away from poverty in old age. Across an entire career, even short periods of wage and returns fluctuations can jeopardize the adequacy of benefits at retirement. Since the outbreak of the coronavirus pandemic in 2020, many people have learned that these risks can lead to temporary months of unemployment, wage reduction, and high market volatility. The heavy reliance of funded accumulations on wage income amplifies these risks.

We find the minimum pension guarantee most effective not only to alleviate poverty and income inequality but also to diversify the above-mentioned risks. This implies placing more weight on the public pillar and balancing the funded and the unfunded pension pillars. In parallel, we suggest increasing savings independently from mandatory contributions to the third pillar. Consistently, the individual must acquire both the knowledge and tools to

manage their funded portfolio during years of savings and retirement. Consequently, any government intervention must first consider encouraging financial education, especially regarding the shift to more funded pension schemes.

Pension schemes with high market exposure are expected to suffer the most from financial and labor risks. As obtained in the literature: "Relying solely on the funded pillar means neglecting the possibility of risks such as volatility of returns, fluctuations in the labor market, and the emergence of systemic risks from time to time" (Barr and Diamond 2009, p. 12).

As time pass from origin reforms executed, we invite further empirical research to examine detailed aspects of the future pension landscape in mixed pension designs with dominant funded pillar. Stress test of systemic shocks must be any part of any examination, especially in the current era of frequent turmoil.

**Author Contributions:** Conceptualization, I.W. and S.L.; methodology, I.W. and S.L.; software, I.W.; validation, I.W. and S.L.; formal analysis, I.W. and S.L.; investigation, I.W. and S.L.; resources, I.W. and S.L.; data curation, I.W. and S.L. All authors have read and agreed to the published version of the manuscript.

**Funding:** This research received no external funding.

**Institutional Review Board Statement:** Not applicable.

**Conflicts of Interest:** The authors declare no conflict of interest.

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
