# Peer review of "Vague Pension Future: Empirical Evidence from the Israeli Radical Privatized Market"

_jrfm, doi:10.3390/jrfm15050207_

Round 1

Reviewer 1 Report

The abstract demands substantive revision so as to accurately summarize the contents of the article (topic, current beliefs, methodology and conclusion). Quoted passages require page(s). Too many non-peer reviewed sources. The manuscript has a low integrative value in the current research. Figures are unclear and need updating. The paper requires major revisions to contextualize the merits of the study and potential uses of its methodology in future studies. The manuscript will benefit from further discussion of key concepts and methodological criteria in order to offer a better articulation between theory and data. There is a need of structuring the discussion to ensure that the methodological aspects are clearly presented. Many statements appear in the discussion section without explanation as to the data on which they are based. ‘Given strong American influence’ – U.S. would be better. The reference list is poorly edited.

Author Response

Many thanks for your contribution to this manuscript. attaching a word file explaining the corrections and modifications made. please find above the correct manuscript.

Reviewer 2 Report

The authors examine an important question. The research is well implemented. Overall, well done!

Author Response

Thanks for your warm words. I appreciate your review.

We shortened the Introduction and made the discussion more in focus explaining the methodological process. in addition we improved dome spelling and grammar falls.

Many thanks

Reviewer 3 Report

This paper projects the future benefits resulting from the Israeli pension system and makes assessments regarding the adequacy of benefits and the need for a minimum pension. Given the nature of the highly privatized pension system in Israel, they argue for more Government intervention in order to address issues like income inequality and providing a minimum standard of living for the retirees. They also point to other downsides of a highly privatized pension system.

Some possible suggestions and questions:

  • Table 1 is in national currency. It would be more relevant to have it scaled like with replacement rates. In fact, the authors do comment on replacement rates and they should be provided for all categories.
  • The authors should comment more on the subject of adequacy. How can we define adequacy? References could be used.
  • The authors find that for some groups the benefits are small. Is it because of low saving/insufficient time in the system/other reason? The whole paragraph with adequacy should be improved as there isn’t much clarity.
  • Somehow from the paper one could get the idea that insufficient benefits at pension are a result of a privatized system. Is there the case or simply there is insufficient saving? Do the author argue for a higher share of a PAYG system?
  • The authors could insist more on the income inequality generated by such a privatized system.
  • The part related to the minimum pension guarantee is insufficiently developed. The authors claim that it would benefit redistribution. By how much? Do you have simulations about the impact? More proofs and calculations for that topic would be needed.

Overall, the main empirical work in the paper is related to the calculation of future pension benefits for a sample of Israeli future pensioners. A solid justification of how these calculations contribute enough to the dedicated literature is needed.  

Author Response

Thank you for your connemts

Here I attach a rebuttal letter to clarify the corrections. in addition, you will find the revised manuscript above.

Thanks the Authors

Round 2

Reviewer 1 Report

The revised version can be published.

Reviewer 3 Report

The revised version of the paper contains only limited changes related to the core manuscript. In the first round of review, I raised several issues but these have not been addressed, either in the paper or with comprehensive arguments. Largely, I have the main concerns and observations from the first round of review.

Related to table 1, my suggestion was about scaling the results, for example with replacement rates.

Overall, I don’t think that the manuscript has sufficiently improved to warrant publication. In addition, the core question about the way in which the calculations performed by the authors contribute to the literature still remains.